# Preliminary Study on Assessing Delaminated Cracks in Cement Asphalt Mortar Layer of High-Speed Rail Track Using Traditional and Normalized Impact–Echo Methods

**DOI:** 10.3390/s20113022

**Published:** 2020-05-26

**Authors:** Ying-Tzu Ke, Chia-Chi Cheng, Yung-Chiang Lin, Yi-Qing Ni, Keng-Tsang Hsu, Tai-Tung Wai

**Affiliations:** 1Department of Construction Engineering, Chaoyang University of Technology, Wufeng 413310, Taiwan; yingtzu0125@gmail.com (Y.-T.K.); yungchaing.lin@gmail.com (Y.-C.L.); frankhsu0423@gmail.com (K.-T.H.); 2Department of Civil and Environmental Engineering, The Hong Kong Polytechnic University, Hun Hom, Kowloon, Hong Kong, China; yiqing.ni@polyu.edu.hk (Y.-Q.N.); tai-tung.wai@polyu.edu.hk (T.-T.W.)

**Keywords:** impact–echo, normalized amplitude spectrum, cement asphalt mortar, high-speed rail track slab, crack, concrete, flaw detection

## Abstract

The severe deterioration of a cement asphalt (CA) mortar layer may lead to the movement of the upper concrete slab and impair the safety of the speedy train. In this study, a test specimen simulating the structure of high-speed rail track slabs was embedded with delaminated cracks in various lateral sizes inside the CA mortar layer. Impact–echo tests (IE) were performed above the flawed and flawless locations. In present study, the IE method is chosen to assess defects in the CA mortar layer. Both traditional IE and normalized IE are used for data interpolation. The normalized IE are the simulated transfer function of the original IE response. The peak amplitudes in the normalized amplitude spectrum and the peak frequency in the traditional amplitude spectrum for the top concrete overlay were used to develop simple indicators for identifying the integrity of the CA mortar layer. The index was based on the difference of the experimental peak amplitude and frequency of the ones calculated from previously developed formulas for plates without substrates. As a result, the technique does not require an experimental baseline for the crack assessment. A field test and analysis procedure for evaluating high-speed rail slab systems are proposed.

## 1. Introduction

The cement–asphalt–mortar layer (CA mortar layer) is located between the upper pre-stressed concrete track slab and the base concrete roadbed. After service for many years, cracks form inside the CA mortar layer and show on the side surface. As shown in Figure 1, debonding of CA mortar is one of the main damage types in China’s railway track system II slab ballastless track [1]. Thermal deformation may be the cause of the interfacial separations [2,3,4,5,6]. Severe deterioration of the CA mortar layer may lead to movement of the upper concrete slab and compromise the safety of the speedy train. Finding delaminated cracks inside the CA mortar layer by a non-destructive method is not a simple issue. The problem cannot be assessed by infrared as the concrete slab above the CA mortar layer is thick—typically 0.2 m. Ground penetration radar (GPR) is not a suitable choice either, as this method is not sensitive to crack with small opening width [7,8]. Although structural cracks in concrete are detectable by the stress wave methods such as ultrasonic echo [9] or the impact–echo method (IE) [10,11], identifying cracks inside the CA mortar layer is rather difficult. As CA mortar has much lower acoustic impedance than concrete, for ultrasonic-echo methods, strong reflecting signals can be obtained in the time domain from the concrete–CA mortar interface as well as the concrete–crack interface. For the IE method, the dominant peak frequency in the amplitude spectrum corresponding to the concrete layer with or without substrates with lower acoustic impedance is similar [12]. The Spectral Analysis of Surface Waves (SASW) and Multi-channel Analysis of Surface Waves (MASW) methods are mainly chosen for assessing the layering thickness and properties [13,14,15,16,17,18,19]. However, the application of these methods usually aims at assessing concrete with surface damages or with layer variation [20] and is more time consuming than the other conventional Non-destructed test (NDT) methods.

In present study, the IE method was chosen to assess defects in the CA mortar layer. Both traditional IE and normalized IE [21,22,23,24,25] were used for data interpolation. The traditional IE method evaluates the thickness of a plate or depth of a defect by the high amplitude peak frequencies in the spectral domain which is produced by S1 modal vibration of the plate-like structure or the stress waves propagate back and forth between top surface and boundary of defects [9,26]. The peak amplitude may alter with the change of the impact–force amplitude and the hitting duration of the steel ball. In contrast, a transfer function is the spectrum corresponding to an impulse that applies the same energy to the test object across all frequencies, so the amplitude of the high-amplitude peak corresponding to the wave reflection from the interface is determined only by the acoustic conditions on the interface [26,27]. The time history of the impact force must be known to determine the transfer function from an IE response. In order to record the force–time function, it is necessary to use an impactor with an embedded sensor. As the diameter of the impactor applied in IE is usually between 3–10 mm, the sensor is hard to be implanted and to be calibrated. Moreover, different testing targets require different diameters of the striking source. The instrumentation to obtain transfer function is much more expansive than the regular IE. Therefore, a normalized IE solely obtained from the IE displacement response to simulate transfer function is proposed. In the simulated transfer function, the modified Rayleigh wave (R wave) displacement obtained from the displacement waveform of traditional IE is used as the force–time function. After modification, the amplitude of the R-wave has the magnitude proportional to the force magnitude and with the same duration. The previous studies show the peak amplitude of S1 modal vibration of a plate is fairly constant regardless of the force-magnitude and duration [22]. A clear descending trend of the overlay thickness–amplitude with respect to the increase of the ratio of acoustic impedance of overlay and substrate was found in the simulated transfer function for the models with semi-infinite substrate [23]. The thickness–amplitudes of the concrete overlay for substrate layers made by epoxy, low-strength mortar and gypsum are substantially lower than the plate response [23].

In this study, the IE experiments were conducted on a concrete–CA mortar–concrete sandwiched slab specimen. The CA mortar layer contains different sizes of delaminated artificial defects at the top or bottom of the layer. The theoretical background of the normalized IE analysis is introduced first, followed by the description of the experimental design. In the result section, both traditional IE and normalized IE spectra for all the scenario are demonstrated first from the representative cases. Indices are then introduced to demonstrate the statistical characteristics of the test results for different bonding situations. In the section of discussion and conclusion, the pros and cons of using the traditional IE and normalized IE are discussed. The advantage of using one index to identify defects within the CA mortar layer is also explained. A field test and analysis procedure for evaluating high-speed rail slab systems is proposed.

## 2. Theoretical Backgrounds

### 2.1. IE Response for a Plate

In the method, transient stress waves are introduced into a structure by mechanical impact produced by tapping a steel ball at a point on the surface. A displacement transducer located close to the impact point monitors the vertical surface displacements. The displacement waveform generated by the impact on a plate-like structure is shown in Figure 2a. The dominant response after the arrival of the Rayleigh wave (R-wave) corresponds to the transient vibration of the S1 Lamb wave mode, which acts as the local contracting and expanding in the thickness direction [14]. After Fast Fourier Transformation (FFT), this periodical excitement produces a dominant peak in the amplitude spectrum shown in Figure 2b. The peak frequency, which is named as the thickness frequency, can be calculated by Equation (1) with a known *P*-wave speed of concrete (*C_p_*) and the thickness of the plate (*T*) [10]. For a composite plate with n layers, the thickness frequency can be estimated by the inverse of the back and forth traveling time between two plate surfaces for *P*-wave with known speed(*C_pi_*) and thickness (*T_i_*) as shown in Equation (2) [10].
(1)$$f=\frac{0.96C_{p}}{2T}$$
(2)$$f=\frac{1}{\sum_{1}^{n}\frac{2T_{i}}{0.96C_{Pi}}}$$

### 2.2. Normalized IE Spectrum 

The normalized IE spectrum is a simulated transfer function of the original IE response. Instead of using the force–time function produced by the impactor to obtain the transfer function, the modified Rayleigh wave (R-wave), which is the first arrived dominant displacement waveform shown as point A to C in Figure 3a, is served as the simulated force–time function [21,22,23]. The duration of the Rayleigh wave, which is the time span between A and C in current example, should be identical to the duration of the force–time function obtained from the impactor, as the R-wave is non-dispersive while traveling in a homogeneous elastic medium [24]. However, the ratio between the magnitude of the R-wave and the force–time function varies with the impact-duration and the impactor–receiver distance. In order to use the R-wave as the simulated force–time function, a constant ratio between the two magnitude need to be achieved. Thus, the theoretical relationship between the maximum magnitudes of the R-wave displacement, A_peak_, with the known impact force durations (t_d_) and impactor–receiver distances (r) and the maximum displacement corresponding to the step-functional force with the same magnitude as the maximum force magnitude in the impact–echo force–time function were derived as shown in Equation (3) [24]. In Equation (3) the dimensionless impact-duration, τ_d,_ which equal to C_s_t_d_/r, containing the information of material properties of the media.
(3)$$\{\begin{array}{l} y=f(x)=x^{-\frac{1}{2}}\\ y=f(x)=1+0.12x^{-1.7}\\ y=f(x)=1 \end{array}\;\mathrm{for}\;\begin{array}{l} x\leq 0.4\\ 0.4<x<2.5\\ x\geq 2.5 \end{array}$$
where y = A_peak_/A_0_, x = τ_d_/γ; γ is the dimensionless arrival time for R-wave w.r.t. the arrival time of S-wave, 1.1018 for the present case; τ_d_ is the dimensionless impact-duration, C_s_t_d_/r; C_s_ is the shear velocity of concrete.

Thus, we proposed a simulated force–time function, which is proportional to the force magnitude regardless of the impact-duration, the distance to the exciting force and the material type, is the R-wave divided by a factor, *F_n_*, provided in Equation (4) [24], where we let *r* equal to 0.03 m and the *P*-wave speed, *C_p_*, equal to 4000 m/s as the default values, then *F_n_* for arbitrary *r* and *C_p_* is
(4)$$F_{n}=y\times {\left(\frac{4000}{C_{p}}\right)}^{2}\times \left(\frac{0.03}{r}\right)$$

Figure 3 presents the procedure for calculating the normalized spectrum from an IE displacement waveform. Figure 3a is the IE displacement wave form. The duration of the R-wave, td, is defined by the time window between points A and C in Figure 3a, where point A is the first crest of the R-wave and point C is the time for the displacement return to zero. Figure 3b shows the corresponding amplitude spectrum. Figure 3c is the displacement of the R-wave extracted from Figure 3a. The simulated force–spectrum is obtained by performing FFT on the R-wave displacement divided by Fn. The normalized amplitude spectrum, as shown in Figure 3e, is the amplitude of the IE spectrum shown by Figure 3b divided by the amplitude of the simulated force spectrum shown by Figure 3d. It needs to be noticed that in the normalized amplitude spectrum, the amplitude for the frequency of over 1.25/td may be falsely enlarged because of the amplitude of simulated force spectrum, such as shown in Figure 3d, is near zero beyond this frequency threshold.

### 2.3. Thickness–Amplitude of Concrete Plate-Like Structure without Substrate

A regression formula, shown in Equation (5), has been developed to estimate the amplitude of the peak at the thickness frequency, called by thickness–amplitude, in the normalized spectrum for homogeneous concrete plates [22]. In Equation (5), the thickness–amplitude is computed from given values of plate-thickness (T, in mm), the ratio of impactor–receiver distance to thickness (*r*/*T*) and *P*-wave speed (*C_p_*, in m/s). The thickness–amplitude is dimensionless as it is the ratio of point-wise dividing between the amplitude spectra obtained from the displacement waveforms of the IE and the Rayleigh wave responses multiplied by a factor $F_{n}$. The formula at the right of the Equation (5) is fitted from the numerical simulations with *T*, *r* and $C_{p}$ as the variables [11].
(5)$$thickness-amplitude=(\frac{1097.5\times {\left(r/T\right)}^{-0.3764}}{T^{1.25}})\times {\left(\frac{4000}{C_{p}}\right)}^{2}$$

The previous research shows the deviations between the average experimental and computed thickness–amplitude are mostly within 10% for the impact-duration, which can be estimated by the duration of the R-wave, less than 30 μs [23].

## 3. Experimental Design

### 3.1. Specimen Design

Figure 4 shows the schematic layout of the test specimen. The two concrete plates sandwiched with a CA mortar layer at the center. The mixture of the CA mortar is the same as the one in the CRTS II slab ballastless track. The thickness of the upper concrete slab is 250 mm. As shown in Figure 4, two layers of grid steel reinforcement with a diameter of 10 mm are placed in the upper concrete slab, and one layer is placed in the lower concrete slab to improve the bending capacity of the lifting. The cover thickness of the steel bars is 50 mm. The specimen was divided in three zones labeled as A, B and C. The width of Zones A, B, C are 500 mm, 500 mm and 400 mm, respectively. The CA mortar layer is about 35–38 mm for Zones A and C and is about 135 mm for Zone B. The former CA layer thickness is close to the designed thickness for the high-speed rail track for China’s Beijing-Shanghai high-speed railway 30 mm [28]. The latter one is the extreme case considering bad controlling in layer thickness. As a consequence, the thickness of the lower concrete slab is 260 mm in Zones A and C, and 155 mm in the Zone B.

Thin Styrofoam sheets, 2 mm in thickness, were placed in the CA layer to simulate the cracks. Three sheets with lateral dimensions 300 mm × 300 mm, 200 mm × 200 mm and 100 mm × 100 mm were placed at each zone shown in Figure 4. In Zone A, the Styrofoam sheets are on the top of the CA mortar layer labeled by A30, A20, A10, respectively. The defects B10 and B30 in Zone B are placed at the bottom of the CA layer and B20 is on the top of the CA layer. The defects in Zone C were positioned at the bottom of the CA mortar layer, except for C20 which floated to near the top of the CA mortar during casting as shown in Figure 5a. Figure 5b shows the side view of the specimen after casting. The IE tests were performed above the center of the defects and the good adhesion regions as shown in Figure 5c.

### 3.2. Instrumentation

As shown by the photo in Figure 6, the instrumentation of the IE test system consists of three components: an impactor, a receiver and a portable computer-based data-acquisition system. The impactors are spherical steel balls with diameters from 3 mm to 20 mm [10]. A broadband displacement receiver comprising a small, conically shaped, piezoelectric element cemented to a brass cylinder is employed to record vertical displacements of a concrete surface. The time interval used to record the waveforms retrieved by the receiver is 1.6 μs in all the tests. The record length is 4096 data points. Therefore, the frequency interval used in digitizing the amplitude spectrum is 0.153 kHz. The diameters of the steel balls were 7 mm, 9 mm and 11 mm in diameter. The distance between the tapping location and the receiver was 50 mm. The tapping positions were either directly above the defects or at the good adhesion regions between the cracks. Three repetitive tests were performed for each test position.

## 4. Experimental Results

### 4.1. Zones A and C

The thickness of the CA mortar layer in Zones A and Zone C is about 36 mm. The defects located on the top and bottom of the CA layer for Zone A and Zone C, respectively, except for C20. As the measured *P*-wave speed of concrete, the impactor–receiver distance and the slab thickness are 4200 m/s, 50 mm and 250 mm, respectively, the expected plate thickness frequency and amplitude of the top concrete slab calculated by Equation (1) is 8.064 kHz, and the expected thickness–amplitude of the top concrete slab obtained from Equation (5) is 1.835. The *P*-wave speed of the CA layer in the present case is 1630 m/s, which is obtained by direct ultrasound measurement of the test cylinder. Including the wave speeds and the thickness of the three layers, the thickness frequency for the whole plate is 3.386 kHz using Equation (2).

For comparison, Figure 7a–d show both the representative IE spectrum and the corresponding normalized amplitude spectrum for the good adhesion, A10, A20 and A30 cases, respectively. The black dot in the normalized spectra indicates the peak close to the calculated thickness–frequency of the concrete overlay, 8.064 kHz. The exact frequency and amplitude are shown on the side of each spectrum by the overlay frequency and amplitude. The empty dot indicates the peak close to the thickness–frequency of the composite plate, 3.386 kHz. The exact frequency and amplitude are shown on the side of each spectrum by the “Bottom” frequency and amplitude.

Using traditional IE methods, we can determine whether the test location is defective by comparing the main peak frequencies in the spectrum of the defect and the good adhesion case. For good adhesion, A10, A20 and A30, the overlay frequencies, shown at the upper part of Figure 7a–d, are 13.6%, 2.8%, 2.2% and 2.2% higher than the calculated thickness–frequency 8.064 kHz, respectively. The frequency difference to the one calculated by Equation (1) is within 3% for the cases containing defects. Regarding to the peak frequencies near the “Bottom” frequency, they are 2.9 kHz, 2.44 kHz and 1.96 kHz for A10, A20 and A30, respectively, which are much lower than the calculated frequencies of 3.386 kHz. The larger lateral dimension of the crack leads to lower the peak frequency. On the other hand, for good adhesion case, the peak frequency is 3.82 kHz, which is fairly close to the calculated frequency.

Obtaining from the normalized IE spectra shown at the lower part of Figure 7a–d, the difference between the overlay amplitudes of good adhesion, A10, A20 and A30 to the one calculated by Equation (5) are −11.4%, −6.3%, 1.7%, 7.1%, respectively. Notice that, as the contact time td is about 40–50 μs, the reliable peak amplitudes are the ones below 25 kHz (1.25/50 μs = 25 kHz). The variation of the normalized amplitudes is within 10% in all the cases with defect. The largest “bottom” amplitude peak is found in the spectrum of the good adhesion case.

Figure 8a–d show the representative traditional and normalized amplitude spectra in Zone C for the good adhesion and C10, C20 and C30 cases, respectively. In Figure 8a, the overlay frequency and amplitude are 9.0 kHz and 1.656, respectively. The results are very similar to the good adhesion case in Zone A. For Figure 8b–d the highest peak frequencies locate at 6.56 kHz, 7.48 kHz and 7.78 kHz, respectively and the amplitudes are 1.531, 1.484 and 1.516, respectively. As for the cases with defects, the waves refracted into the CA layer and diffracted from the defect. The dominant peak amplitudes become multiple and less distinct. Employing the *P*-wave speed of the CA layer in the present case, 1630 m/s, to Equation (2), the frequency for waves traveling back and forth from the top 2 layers is 6.173 kHz, which is close to the lower bound of the multiple peaks shown in the cases with the defects in Zone C. In contrast, the normalized amplitude of the dominant peak for the defect-case is not significantly lower than the good adhesion case. The overlay amplitudes are 1.656, 1.531, 1.484 and 1.516 for good adhesion, C10, C20 and C30, respectively. Regarding to the bottom frequency, only the good adhesion case in Figure 8a shows distinct peak at the frequency close to the calculated thickness frequency of the whole plate. As the stress waves block by the crack, the peak corresponding to the reflection from the bottom of the plate is not apparent in the spectra of C10 and C30. Notice that, in Figure 8a, there is a distinct peak at 20 kHz. This peak is considered to correspond to the intersection of the reinforcing bars for the following reasons. First, the values of overlay frequency, the bottom frequency and the normalized amplitude are similar to the ones shown in good adhesion case. Second, if there is only one rebar below the measuring point, there should be no rebar response as the ratio of the rebar diameter to cover thickness (d/c) is less than 0.3 [12]. At the intersection, the total range of steel bars is 20 mm, so the d/c is larger than 0.3. The rebar response is expected at 21 kHz (4200/4/0.05 = 21,000 Hz). The peak near 1 kHz in Figure 8a–d corresponds to the resonant frequency of the displacement transducer and is not related to the structural scenarios.

In the representative cases, we select the frequency and normalized amplitude corresponding to the first layer and the whole composite plate as the key parameters. *R_amp_* and *R_frq_* are introduced to express the key characteristics in the normalized amplitude spectra as defined by Equations (6) and (7). The *R_amp_* is the rate of the amplitude difference between the experimental and the base amp. calculated by Equation (5), whereas the *R_frq_* is the rate of the frequency difference between the experimental and the base freq. calculated by Equation (1) with known overlay thickness and C_p_. Figure 9a demonstrates the average *R_amp_* and the standard deviation obtaining from all the test data in the experiment. When the defects are located at the bottom of the CA layer, shown as C10, C20 and C30, the average thickness–amplitude of the upper layer decreases to about 18%. As the *R_amp_* for good adhesion cases, labeled as GB, also decreases to about 15% in average, it is hard to identify the difference by *R_amp_* when the defects are on the bottom of CA layer. On the other hand, for defect locating on the upper surface of the CA layer, shown as A10, A20 and A30, the average *R_amp_* increases with the lateral dimension of the defect. For A20 and A30, the lower limit of *R_amp_* is greater than −10%, which is similar to the results obtained from concrete slabs without a substrate. Therefore, defects with a lateral dimension equal to or greater than 0.2 m can be distinguished from flawless cases.
(6)$$R_{amp}=\frac{Exp.Amp.-base\;Amp.}{base\;Amp.}\times 100\%$$
(7)$$R_{frq}=\frac{Exp.freq.-base\;freq.}{base\;freq.}\times 100\%$$

Figure 9b shows the *R_frq_* of all experiments performed above defects and good adhesion areas. Unlike the distribution of normalized amplitude, the peak frequency variations are limited for each case. The lowest *R_frq_* obtained from the good adhesion cases is 8%, whereas largest *R_frq_* is about 5% for A10 and A30. For C10, C20 and C30, the *R_frq_* are always negative. Therefore, using only the highest frequency as a parameter for defect identification, a horizontal defect located near the top surface of the CA mortar layer may be misjudged by slightly change in thickness of the concrete overlay. A defect near the bottom of a CA mortar layer can be more accurately identified as the large of the frequency.

A parameter, AF-index, is proposed to assess defects more accurately and quickly within 36 mm CA mortar layer without knowing the position of the defect. The AF-index is defined by the multiplication of *R_frq_* and *R_amp_*, as shown in Equation (8). Figure 10 shows the relationship between the AF-index and the lateral dimension of defects regardless of the location of the defect within the CA mortar layer. For the good adhesion cases, the AF-indices are mostly below −50. For defects on the top of the CA layer, the AF-indices are mostly between −50 and +50. For the defects on the bottom CA layer, the AF-indices are mostly larger than 50 and decrease with the increase of the lateral dimension. Notice that. as the defect of C20 is floated from the bottom, the AF-index are close to the ones of A20. Thus, it is reasonable to conclude that a defect with the lateral size larger than 0.1 m may be presented within the CA layer when the AF-index is larger than −50.
(8)$$\mathrm{AF-index}=R_{amp}\times R_{frq}$$

Another strong indicator for identify the good adhesion case is the peak near the thickness–frequency of the composite plate calculated by Equation (2). When there is no defect within the CA mortar layer, the corresponding peak is always strong and very close to the calculated thickness–frequency. Figure 11 shows the diagram of *RB_frq_* versus corresponding peak amplitude obtaining from all the test spectra, where the *RB_frq_* is defined as the rate of the frequency difference between the experimental and the calculated thickness–frequency of the composite plate similar to Equation (6). The good adhesion spectra shown in Figure 7a and Figure 8a have more distinct peak with the frequency close to the calculated frequency of the composite plate. Thus, in Figure 11 using the *RB_frq_* equal to 0 and normalized amplitude equal to 1 as the division lines, most of the good adhesion cases are distributed at the first quadrant.

### 4.2. Zone B with the CA Mortar Layer of 135 mm

In Zone B, the thickness of the CA layer is 135 mm. Both the defects labeled B10 and B30 are at the bottom and B20 is on the top of the CA layer. Figure 12a–d show the representative results. Since the peak frequency corresponding to the concrete overlay for all the representative cases is identical, 7.78 kHz, the traditional IE method cannot assess defect within CA mortar layer in Zone B. The normalized amplitudes are 3.32, 2.786, 2.653 and 1.89 for good adhesion and B10, B20 and B30 regions, respectively. The amplitudes for B10 and B20 are slightly lower than the good adhesion case, and the amplitude is much lower for B30.

Figure 13 shows the relationship between of normalized amplitude corresponding to the first concrete layer and the lateral size of the defect for all the tests performed on the region containing the 135 mm CA mortar layer. The minor vertical axis indicates the $R_{amp}$ calculated from Equation (5) using the same *base Amp.* for Zones A and C. For most of the good adhesion cases the $R_{amp}$ is above 50% and defective case is below 50%. Moreover, for the test results for defect with lateral size 0.3 m, the $R_{amp}$ spreads between −30% to 40% and the average $R_{amp}$ is close 0, which means the average amplitude is close to the predicted amplitude for single-layered concrete plate. However, for all the cases, the dominant peak frequency is 7.78 kHz. The peak corresponding to vibrations of the whole thickness of the composite plate is absent in all the IE spectra.

## 5. Discussion and Conclusions

The feasibility of using the traditional IE and normalized IE spectrum to identify the delaminated flaw within the CA mortar layer in China high-speed rail track slab system was explored by performing experiments on the specimen with artificial cracks embedded in the upper surface or lower surface of the CA-layer of a composite plate. The composite plate is composed of two concrete plates sandwiched with the CA mortar layers with a thickness of 36 mm or 135 mm.
For a track slab with a 36-mm-thick CA mortar layer:From the perspective of traditional IE, for the cases of good adhesion and peeling of the top concrete-CA layer interface, the thickness frequencies of the first concrete layer are all close to or higher than the value calculated by Equation (1) and decrease with the increase of the size of the defect. The higher thickness-frequency for the good bond case is caused by the higher cut-off frequency of the lamb wave mode excited by the impact-echo method [14]. Figure 14 shows the theoretical dispersion curves and the corresponding mode shapes for the S1 mode of a concrete slab and the ones for the 6th mode of the three layered composite plates. The dimensions and the material properties of the composite plate are the same as the present specimen. The one-layered concrete slab can be considered as the totally debonding situation. In Figure 14, the mode shapes for the slab and the top layer of the composite plate are very similar. The cut-off frequencies of the modes, which can be referred to as the thickness-frequency in the impact-echo spectrum, are 8.39 kHz for single-layered plate the 9.04 kHz for the three-layered plate [29]. In comparison, the thickness frequency shown in Figure 7d for A30 and Figure 7a for good bond case are 8.24 and 9.16 kHz, respectively. Although the peak frequency is slightly lower for the case with a defect, there may be local variations in the thickness of the concrete cover layer on-site, so the situation with defects above the CA layer may be mistaken for a good adhesion. For the cases where the defect is deeper in the CA mortar layer, the peak corresponding to the wave reflections between the defect and concrete surface is substantially lower than the one calculated by Equation (1) because the waves refracted into the CA layer and diffracted from the defect. This type of defect can be easily evaluated using the traditional IE. The thickness–frequency corresponding to the whole thickness of the composite plate decreases with the presence of a defect in the CA mortar layer because of the longer route for the wave propagation with the presence of the defect. However, this parameter may not be reliable for on-site inspection as the peak corresponding to the whole composite plate may not be obvious when the track plate is closely bonded with the foundation and the decreases in frequency may due to flaws in the concrete plates.From the perspective of the normalized IE, the normalized thickness amplitude of the case with the upper interfacial defect more than 0.2 m in size is mostly higher than the one with good adhesion and the lower interfacial defect and close to the one calculated by Equation (5). The dominant peak amplitudes become multiple and less distinct for the cases with defects on the bottom of CA mortar layer and with good adhesion because the waves dissipate into the CA layer. Because the traditional IE has the advantage of detecting the cracks on the lower surface of the CA mortar layer, and the normalized IE has the advantage of detecting the upper surface defects, in order to quickly and systematically estimate the defect status, the AF index is introduced. The AF-index is the product of the increase rates of the peak frequency and the increase rates of the normalized amplitude. The baseline frequency and amplitude can be calculated with the known upper concrete plate thickness, impactor and receiver spacing and concrete P wave velocity. For most well-bonded cases, the AF index is less than −50. For the defects with a lateral dimension equal to or greater than 0.1 m, AF is higher than −50 regardless of its position in the CA mortar layer.For the CA layer with a thickness of 135 mm: Traditional IE is not suitable for detection in this case, because the peak frequency corresponding to the concrete overlay is the same for the good adhesion and the defect cases even for the case with 0.2 m defect locate on the top of CA mortar layer. Moreover, the peak corresponding to the whole thickness of the composite plate is missing as the stress waves cannot penetrate the thick CA mortar layer. However, the normalized amplitude of the dominant peak corresponding to the concrete overlay may be used as a parameter to identify the existence of cracks. In our experiment, for most of the good adhesion cases, the peak amplitude is about 60% higher than the baseline amplitude. For the cases with defect in the CA mortar layer with the lateral dimensions of 0.1 or 0.2 m, the peak amplitude is about 40% higher. For defects with a lateral dimension equal to or greater than 0.3 m, the amplitude is close to the baseline amplitude of the top concrete slab. Limited by the dimensions of the specimen, only three defects can be placed in Zone B. Hence, we choose B20 at the top and B10 and B30 at the bottom of the CA layer to obtain a broad picture of the characteristic response for the case with a defect inside the 135-mm-thick CA layer. Therefore, more rigorous experiments should be conducted in the future for the composite plate containing thick-CA mortar layer.There is a great chance that the thickness of CA mortar layer for the China railway track system falls within the range between 36 mm and 135 mm. According to this study, due to the change in the thickness of the CA mortar layer, there is a large difference in the characteristics of the IE amplitude spectrum. For the slab containing thicker CA mortar layer, there may not be a peak corresponding to the entire composite slab in the amplitude spectrum, so it is best to use only the main peak frequencies and normalized amplitudes related to the concrete overlay to evaluate defects in the mortar layer. The AF-index is a reliable indicator for slab with thinner mortar layer. For slabs with thick mortar layers, the change in the normalized amplitude of the main peak corresponding to the concrete overlay is the only indicator for defect assessment.

Based on the finding in the present study, a procedure for the field test and analysis for assessing the high-speed rail track slab system with the CA layer-thickness close to the designed thickness is proposed as following.
Measuring the geometric parameters, such as the thickness of the concrete overlay, the CA mortar layer and the bottom concrete slab.Draw grid test points or sampling overlay surface, with the impactor–receiver distance 0.05 m. All the test points should be 0.2 m away from the side edge.Measuring the *P*-wave speed of the concrete overlay. Calculate the baseline frequency following Equation (1) and the baseline amplitude following Equation (5) with known thickness and *P*-wave speed of the concrete overlay.Calculate the baseline frequency of the composite plate following Equation (2) with the *P*-wave speed of the CA mortar layer equal to 1650 m/s as, according to the experience, *C_p_* of CA mortar is usually between 1600–1700 m/s.Performing the IE tests on the test points and obtain the normalized amplitude spectra.Record the amplitudes and frequencies corresponding to the peak near the baseline frequency of the concrete overlay obtained from step 3 and the peak near the baseline frequency of the composite plate obtained from step 4.Calculate the *R_amp_*, *R_frq_*, *RB_frq_* and AF-index of the composite plate to assess the condition of the CA layer and perform the defect assessment within the CA mortar layer.

## Figures and Tables

**Figure 1 sensors-20-03022-f001:**
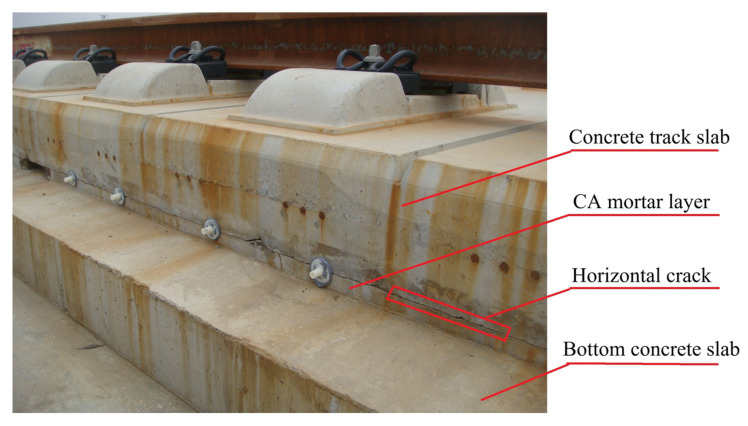
Horizontal cracks show on the side of cement asphalt mortar layer.

**Figure 2 sensors-20-03022-f002:**
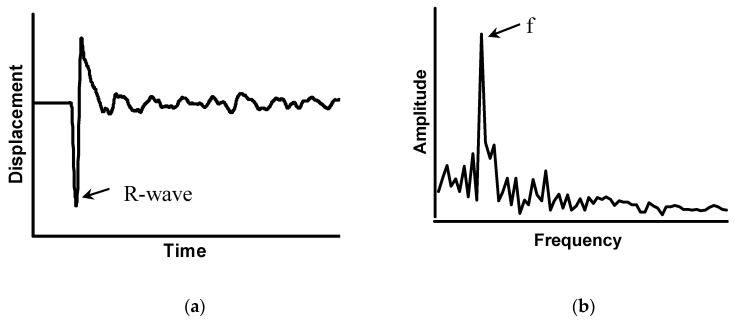
Representative (**a**) displacement waveform and (**b**) the amplitude spectrum for an impact–echo (IE) test performed on a plate.

**Figure 3 sensors-20-03022-f003:**
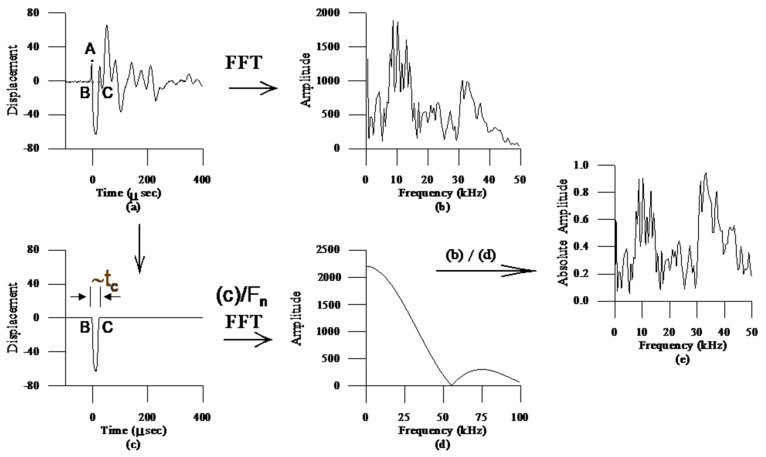
Procedure to obtain the normalized IE spectrum. (**a**) IE displacement waveform; (**b**) IE amplitude spectrum, (**c**) R-wave displacement waveform; (**d**) Amplitude spectrum of R-wave; (**e**) normalized amplitude spectrum.

**Figure 4 sensors-20-03022-f004:**
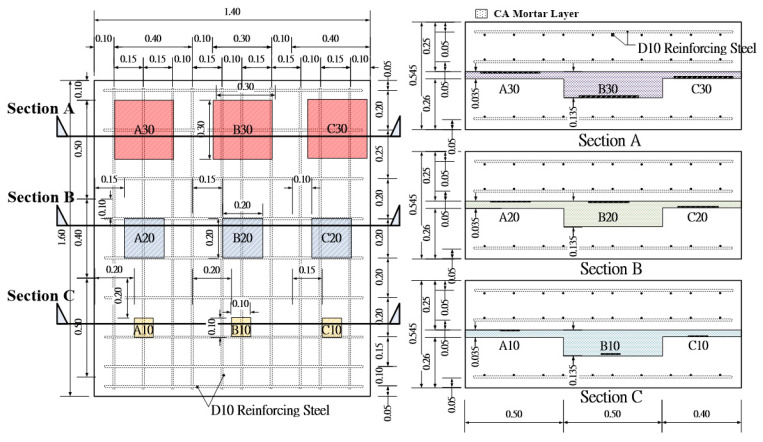
The layout of the specimen design.

**Figure 5 sensors-20-03022-f005:**
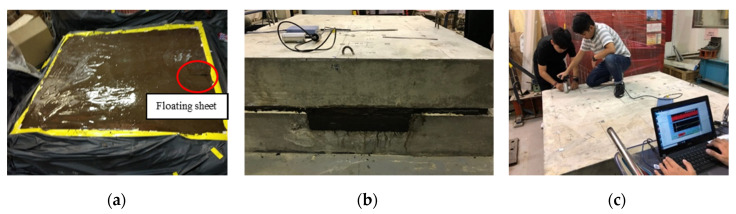
Specimen (**a**) during casting; (**b**) side view; (**c**) the IE test.

**Figure 6 sensors-20-03022-f006:**
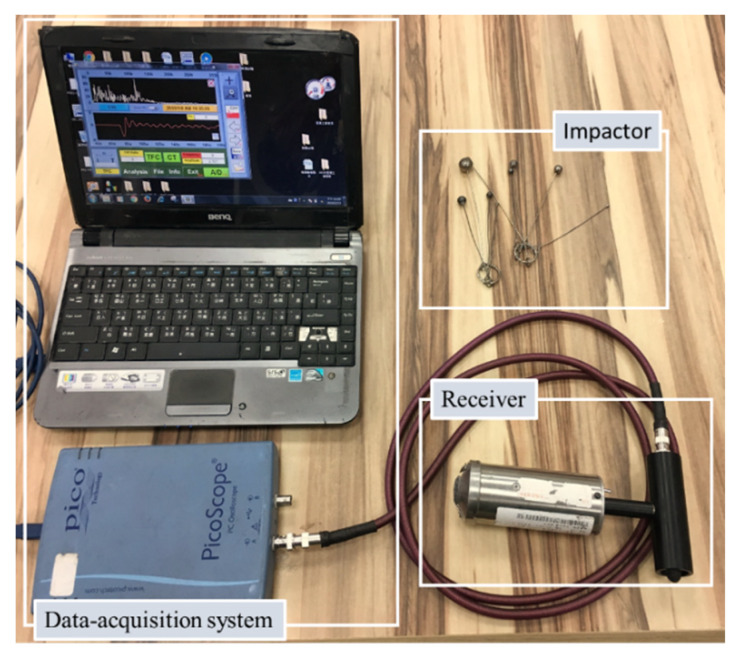
Instruments for performing the IE test.

**Figure 7 sensors-20-03022-f007:**
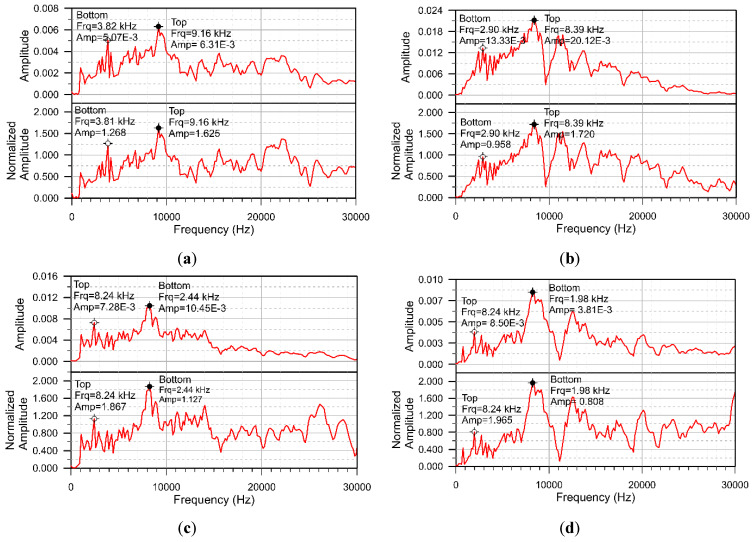
Representative displacement waveform and the normalized amplitude spectra for (**a**) good adhesion; (**b**) A10; (**c**) A20; (**d**) A30.

**Figure 8 sensors-20-03022-f008:**
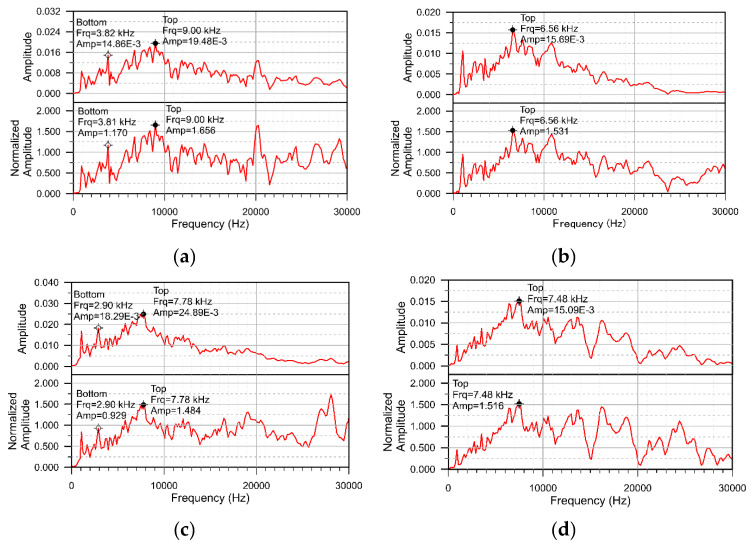
Representative displacement waveform and the normalized amplitude spectra for (**a**) good adhesion; (**b**) C10; (**c**) C20; (**d**) C30.

**Figure 9 sensors-20-03022-f009:**
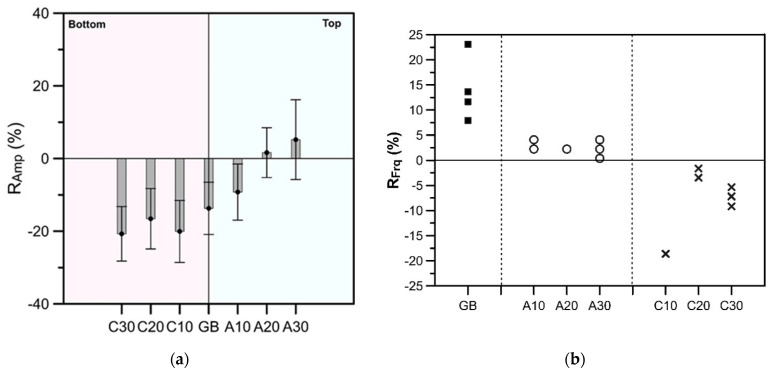
Variation of (**a**) *R_amp_* and (**b**) *R_frq_* for the flawed and the good adhesion cases.

**Figure 10 sensors-20-03022-f010:**
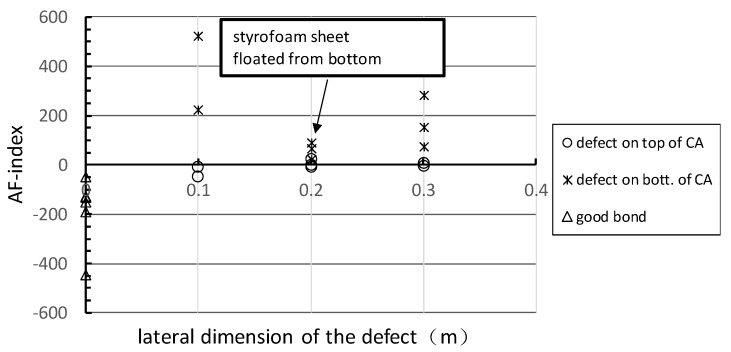
The relationship between the AF index and the lateral dimension of defects for good adhesion, defect on the top and on the bottom of the CA mortar layer.

**Figure 11 sensors-20-03022-f011:**
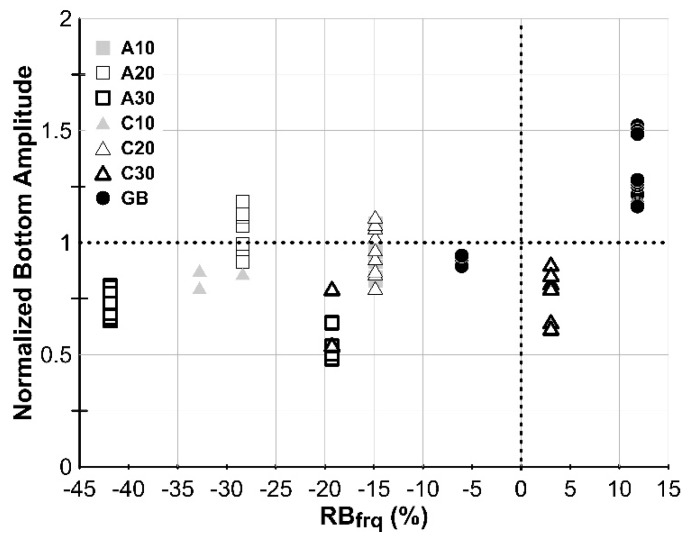
Relationship between the *RB_frq_* and the normalized peak amplitude at the frequency close to the thickness–frequency of the composite plate.

**Figure 12 sensors-20-03022-f012:**
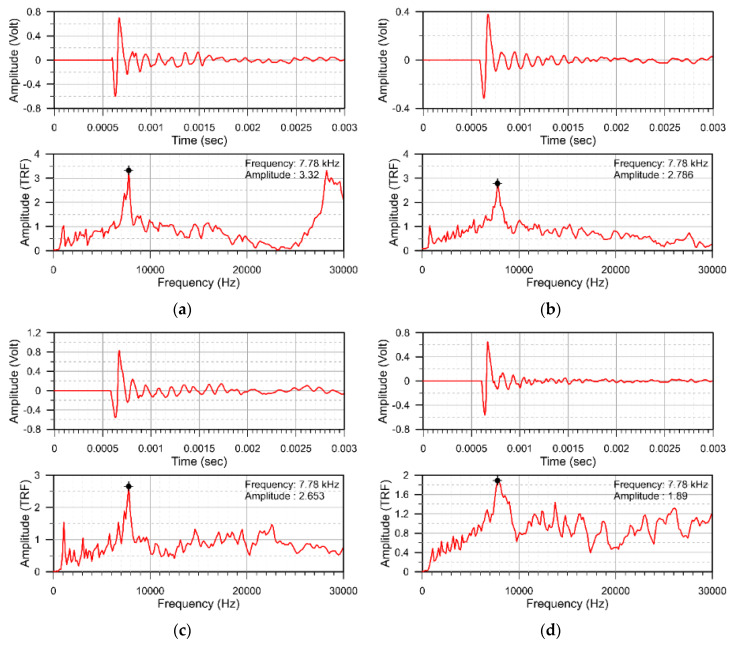
Representative displacement waveform and normalized amplitude spectra for (**a**) good adhesion, (**b**) B10, (**c**) B20, (**d**) B30.

**Figure 13 sensors-20-03022-f013:**
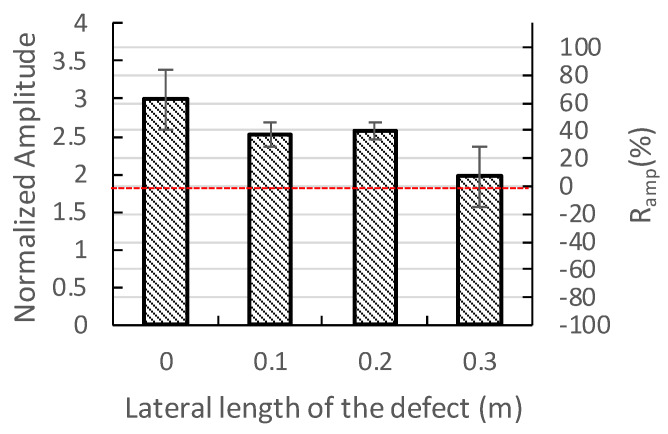
Relationship of the normalized amplitude and the lateral size of the defect for the region containing a 135-mm CA mortar layer.

**Figure 14 sensors-20-03022-f014:**
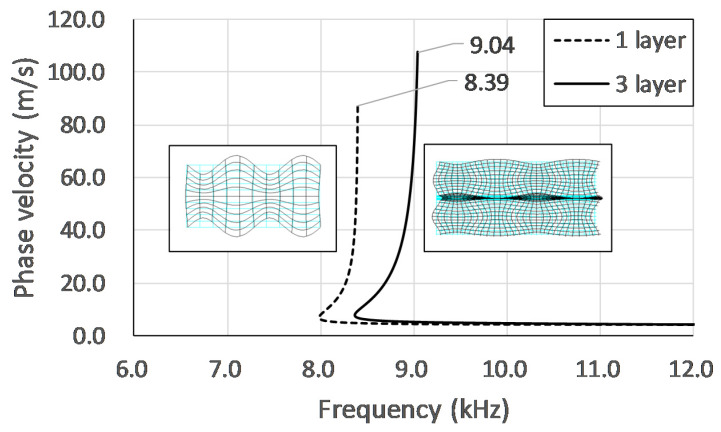
Theoretical dispersion curves and the corresponding mode shapes for the S1 mode of the one-layered slab and for the 6th mode of the three-layered composite plates with the same top layer thickness.

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
