# Peer review of "Preliminary Study on Assessing Delaminated Cracks in Cement Asphalt Mortar Layer of High-Speed Rail Track Using Traditional and Normalized Impact–Echo Methods"

_sensors, 2020, doi:10.3390/s20113022_

Round 1

Reviewer 1 Report

The paper improve a lot. My comments are as follows,

  • (4) does not hold since the dimension of right and left sides of Eq. (4) are different.
  • In Fig. 3, the position of B10 is wrong since “The defects B10 and B30 in Zone B are placed at the bottom of the CA layer” in Line 170.
  • Your discussion is superficial. You explain just experimental behavior. Please explain why (mechanism) you obtain such experimental behavior (amplitude and frequency) with different defect position based on wave propagation in discussion session. Moreover, please divide discussion and conclusion.

Reviewer 2 Report

The authors presented a method to detect delaminated cracks in cement asphalt layer using Impact Echo (IE) testing.

The authors proposed method uses only the received response from the IE testing by normalizing R-wave. 

  1. In reviewer's understanding, the authors have used the  fact that the first peak in the received response contain the impact with some delay. However, in such case, the reviewer think that the fitted model must be a function of material properties. 
  2. The authors have represented defect using a layer of thin styrofoam sheet. Seems very severe damage. Not sure how this represent the real world phenomenon.
  3. Still the theoretical background on how the method would work is weak. Discussions are not as clear on how false-positives can be identified in the real-world practice. 

Round 2

Reviewer 1 Report

My comments are as follows,

Is Fig. 4 correct ?  Reviewer still does not understand them since “B20 at the top and B10  and B30 at the bottom of the CA layer.”  in your sentence.

However, in Fig. 4, B30 and B20 look same position, where the bottom of the CA layer.

IS the following description correct ? Reviewer thinks B20 is on the bottom of the CA layer.

“B20, where the defect is on the top of the CA mortar layer, the thickness-frequency, 7.8 kHz, is as the ones with no defect and defect on the bottom. "

Fig. 14 needs legend. Reviewer understands the explanation of Fig. 14. Based on Fig. 14, please quantitatively discuss to your results.

Please divide chapter for discussion and conclusions.

Round 3

Reviewer 1 Report

I accept present form.

This manuscript is a resubmission of an earlier submission. The following is a list of the peer review reports and author responses from that submission.

Round 1

Reviewer 1 Report

The manuscript presents results of an experimental study of detection of delamination/debonding in or within a cement asphalt (CA) mortar layer of the structure of high-speed rail track. The approach the authors used is normalized impact echo method. Based on the series of conducted tests on the sections of the slab with known embedded Styrofoam sheets simulating delamination, the authors have identified several criteria for their detection.

The missing part of the manuscript is demonstration of advantages of the normalized IE approach over a traditional IE approach. The authors should also include in the manuscript results based on the traditional analysis of IE data. Especially since the results from the traditional IE approach have a more clear physical interpretation and are, thus, more intuitive. For example, in line 263 the authors state that a defect smaller than 0.2 m may not be found. The reviewer feels that the traditional IE would indicate the presence of delamination. 

There are also several suggestions for improvement and errors that should be fixed:

Lines 54 to 56. Presence of a smaller shallow delamination can be detected by a higher frequency peak, not just a lower frequency flexural mode. Line 109. Reference should be to Figure 2. Figure 2. Individual figures should have a brief description. The same with a few other figures. Photo 1. I am not sure if the journal differentiates figures and photos. It might be just another figure. Figure 5a. The authors might comment on the peak at 20 kHz. Why this should be ignored? And why ignore other significant peaks at higher frequencies? Equations 5, 6 and 7, and Figures 6 and 7 should be placed closer to where they are referred to in the text.

Reviewer 2 Report

Authors proposed Impact Echo testing method to detect the cracks in cement asphalt mortar layer. 

Although the topic is interesting and important significant improvement in the manuscript is needed.

General Comments

1) How good is the proposed method compared to existing method?

2) In depth and clear scientific reasoning on how and how well the method works is missing. For example, in calculating transfer function, measuring input direct may bring exact results. 

3) How the authors conducted the experiment is vague. 

4) In Figure 2, is  (c) or B-C point always downward no matter where the measuring points are? or defects are? Is there any phase shift problem if the first peak is in upward direction? 

Specific comments

1) Expressions, sentences, paragraph should be clearer. For example, lines 48-61 is very confusing with little knowledge contained. 

2) All symbols in equations must be explained. Also, in line 79, b -> B. Italic fonts in equations must be in italic as well in the description (text)

3) w.r.t. is not appropriate for a technical paper. In line 109, Figure 1 must be Figure 2. Please refine throughout the manuscript. 

4) In experiment, intact cases and damaged cases are confusing. For some size, A is a intact case whereas in other sizes C is intact case.. Very very confusing to follow. 

5) Photo 1 may be figure as well. Also instruments in Photo1 are hard to see. 

Reviewer 3 Report

This study is Assessing Delaminated Cracks in Cement Asphalt Mortar Layer Using Normalized Impact-Echo Method. The contents of the paper looks reasonable. My comments are as follows,

What is beta. In Eq. (1).  Does Eq. (2) need Beta ? In line 109-113, Figure 1 should be Figure 2. The reviewer does not understand the dimension of right and left sides of Eq. (4). What does the right Figure of Fig. 3 mean? Please consider providing the section view of each defect, as well. The review does not see the device used in the study in Photo (C). I think the novelty an originality of the study is a little bit low. Please explain the novelty an originality of the study, and develop discussion well.